# Role of Circ-ITCH Gene Polymorphisms and Its Expression in Breast Cancer Susceptibility and Prognosis

**DOI:** 10.3390/diagnostics13122033

**Published:** 2023-06-12

**Authors:** Sara F. Saadawy, Nermin Raafat, Walaa M. Samy, Ahmed Raafat, Aliaa Talaat

**Affiliations:** 1Medical Biochemistry Department, Faculty of Medicine, Zagazig University, Zagazig 44523, Egypt; sfsaadawi@medicine.zu.edu.eg (S.F.S.); wmabdelkader@medicine.zu.edu.eg (W.M.S.); atkamel@medicine.zu.edu.eg (A.T.); 2General Surgery Department, Faculty of Medicine, Zagazig University, Zagazig 44523, Egypt; dr_ahmedraafat@yahoo.com

**Keywords:** breast cancer, circ-ICTH, polymorphism, circ-ITCH in increasing risk of breast cancer

## Abstract

Introduction/Objective: Breast cancer (BC) is the first leading cause of cancer-related mortality in females worldwide. We have investigated the correlation between circ-ITCH gene polymorphisms, circ-ITCH expression, and their effect on β-catenin levels and BC development. Methods: Participants included 62 BC and 62 controls matched in terms of age. The circ-ITCH polymorphisms rs10485505 and rs4911154 were genotyped using whole blood samples. In addition, mRNA expression analysis of circ-ITCH was performed on BC tissues. The β-catenin levels in serum samples were measured using ELISA. Results: The qRT-PCR results demonstrated that circ-ITCH was significantly downregulated in BC compared to normal healthy tissues. The genotype distribution of rs10485505 and rs4911154 were significantly associated with BC risk. For rs10485505, genotype CT and TT were significantly associated with an increased BC risk. In contrast, there was a significant association between rs4911154, genotypes GA and AA, and an increased BC risk. Regarding the rs10485505 genotype, carriers of the T allele frequently have a poor prognosis compared to carriers of the CC genotype. Serum β-catenin in the BC patients’ group was significantly higher than in the control group. The relative expression levels of circ-ITCH were remarkably decreased in the BC samples of patients carrying the A allele at rs4911154 compared to patients with a GG genotype. Conversely, β-catenin protein levels were increased in patients carrying the A allele, while rs10485505 genotype carriers of the CT and TT genotypes showed downregulation of circ-ITCH expression fold compared to the CC genotype. Contrarily, β-catenin levels markedly increased in TT and CT genotypes compared with the CC genotype. Conclusions: Our research showed that the rs10485505 polymorphism (T allele) and the rs4911154 polymorphism (A allele) are associated with the risk and prognosis of BC. This finding may be due to the effect on the level of circ-ITCH mRNA expression in BC tissues as well as the level of β-catenin in BC patients.

## 1. Introduction

Breast cancer is the leading cause of cancer-related mortality in women worldwide, accounting for 24.5% of all diagnosed female malignancies [1]. A complex multistep process involving both environmental and genetic differences lead to the development of BC. Age, obesity, prior benign breast disease, a family history of BC, and a woman’s menstrual and reproductive state are all known risk factors for developing BC [2]. The discovery of biomarkers for breast cancer diagnosis will be crucial in assisting with breast cancer treatment, as the differential change of gene expression profile has always been a hot topic in breast cancer research. Therefore, it is important to investigate the characteristic changes of breast cancer from the level of genes [3]. Adding more potential SNPs will have a significant impact on breast cancer risk estimation and will help for earlier use of therapeutic strategies to lower its mortality rate; this suggests the important role of inherited factors in breast cancer susceptibility [4].

With the advancement of science and technology, noncoding RNAs that cannot encode proteins following transcription have gradually come into focus. Recent research has demonstrated that circRNAs are a type of single-stranded closed circular RNA that exhibit conservative evolution with stable structure and tissue specificity [5].

In addition, circRNA has a variety of biological roles, including functioning as a miRNA sponge, protein interaction, regulation of gene transcription, and translation of novel proteins, which suggest that it plays a significant role in the genesis and progression of cancer [6]. Circular RNA predominantly controls the EGFR family, immune cell differentiation, Wnt, Jun, Notch, and MAPK signalling. Circ-ITCH is located at chromosome 20q11.22 on the human chromosome 20. It resembles the RNA structure of the ITCH gene, which typically consists of one to five exons [7]. Low levels of circ-ITCH expression were first observed in oesophageal cancer, followed by colon, hepatocellular carcinoma, and lung cancer [8]. Circ-ITCH controls malignant tumour growth, invasion, migration, and apoptosis, suggesting its role as a key tumour suppressor [9].

A versatile protein called β-catenin is found on the intracellular side of the cytoplasmic membrane. By connecting cadherins to the actin cytoskeleton, it plays a crucial part in cell-to-cell adhesion. It also plays a crucial part in the transcriptional control of the Wnt signalling pathway. In fact, upon Wnt activation, β-catenin translocates from the membrane to the cytoplasm and finally to the nucleus, where it interacts with transcriptional activators to modify a number of target genes linked to accelerated growth, invasion, and cellular transformation, including c-MYC 2 and cyclin D1 [10]. From previous studies, it was established that there is increased cytoplasmic and nuclear β-catenin expression in primary breast cancer [11,12].

Circ-ITCH acts as a miRNA sponge when interacting with miR-216b, miR-7, miR-214, miR-17, and miR-218; it increases the level of *ITCH*, which is involved in the progress of different cancers via the regulation of the Wnt/beta-catenin pathway [13]. The Wnt/β-catenin pathway plays a significant part in the development of many tumours by controlling some essential components of cellular activities; this is carried out by Dishevelled 2(Dvl 2), an essential scaffold protein, bridging the receptors and downstream signalling elements. Dvl2 must be phosphorylated for Wnt signalling to function properly. This process is essential for controlling its stability and activity. ITCH as a member of the E3 ubiquitin protein ligase family can ubiquitinate and degrade the phosphorylated Dvl2 with subsequent inhibition of the canonical Wnt signalling [8,14,15,16].

The purpose of this study was to evaluate the role of circ-ITCH in the carcinogenesis of BC and the mechanism underlying circ-ITCH. Individual cancer susceptibility may be modified by SNPs (rs10485505 and rs4911154) in the circ-ITCH gene region (Figure 1).

## 2. Materials and Methods

The 62 female patients with BC included in this current study were all cytologically confirmed to have the disease. Patients were selected from the Surgery Department, Faculty of Medicine, Zagazig University. Patients who were metastatic at the time of diagnosis, had a history of any other malignancies, were currently receiving neoadjuvant therapy, and had active infectious or inflammatory diseases were all excluded from this study. For the control group, 62 healthy, age-matched women volunteered for medical examination in the Surgery Department of the Faculty of Medicine at Zagazig University. None of them or their families had a history of cancer. All participants included in this study gave signed informed consent. The Faculty of Medicine at Zagazig University’s Local Ethics Committee approved this study, and all procedures followed the Declaration of Helsinki’s ethical principles.

### 2.1. Blood Sampling

Venous blood samples (5 mL) were withdrawn from all BC patients and controls. Samples were placed in plain tubes. Blood was allowed to clot for 15 min before being centrifuged at 4000× *g* for 10 min to separate the serum. The serum samples were stored at 80 degrees Celsius until ELISA detection of β-catenin levels. Another group of whole-blood samples was collected in EDTA-containing tubes and stored at 80 °C until DNA extraction and genotyping were performed.

### 2.2. Tissue Sampling

Human BC tissues and nearby normal breast tissues were taken from BC patients who underwent surgical resection and kept at 80 °C until the circ-ITCH gene expression was determined.

### 2.3. RNA Extraction and Real-Time Quantitative Polymerase Chain Reaction

Total RNA was extracted from human BC tissues and matched adjacent normal breast tissues using TRIzol reagent (Invitrogen, Waltham, MA, USA). The concentration and purity of extracted RNA were determined using a spectrophotometer. Using the QuantiTect Rev. Transcription Kit (Qiagen, Hilden, Germany), total RNA was reverse-transcribed, as directed by the manufacturer. The StratageneMx3005P-qPCR System was used to measure the relative gene expression of circ-ITCH in 25 μL reaction mixtures containing 50 ng of cDNA, 1 μL of 10 pM of each primer (forward and reverse), and 12.5 μL of TOPreal syberGreen master mix (Enzynomics, Daejeon, Republic of Korea). GAPDH was used as a standard internal control. The cycling conditions were as follows: 95 °C for 2 min first for enzyme activation, then cycling for 40 cycles at 95 °C for 10 s, 58 °C for 45 s, and 70 °C for 30 s. The expression of target gene is relatively quantified via normalization against house-keeping gene (GAPDH) according to the calculation of 2^−ΔΔCT^. In short, Δct was determined as the difference in ct between the target gene and the reference gene, and ΔΔct was then determined as the difference between the Δct of the sample and the average Δct of the control. Lastly, 2^−ΔΔCT^ was used for calculation of the fold change in gene expression [18]. The primer sequences were as follows: circ-ITCH (forward 5′-GCAGAGGCCAACACTGGAA-3′, the reverse 5′-TCCTTGAAGCTGACTACGCTGAG-3′) and GAPDH (forward 5′-CCCTTCATTGACCTCAACTA-3′ and reverse 5′-TGGAAGATGGTGATGGGATT-3′).

### 2.4. DNA Extraction and Genotyping

Following the manufacturer’s instructions, genomic DNA was extracted using a G-spinTM Total DNA Extraction Mini Kit from iNtRON Biotechnology, Inc. (Seongnam-Si, Republic of Korea; Catalogue No. 17046). TaqMan SNP Genotyping Assays (Applied Biosystems, Thermo Fisher Scientific, Foster City, CA, USA; rs4911154 SNP Assay ID C__29372891_10, rs10485505 SNP Assay ID C__30396296_20). The TaqMan genotyping assay mix included two primers for the sequence of interest and two TaqMan^®^ Minor Groove Binder (MGB) probes for allele detection. Target-specific oligonucleotides with VIC^®^ dye (linked to the 5′ end of the Allele 1 probe) and 6FAM™ dye (linked to the 5′ end of the Allele 2 probe) were used in the TaqMan^®^ MGB Probes. The genotyping of the two potential variant alleles at the SNP site in a DNA target sequence was made possible by the inclusion of two probe pairs in each reaction. The presence or absence of an SNP detected via genotyping assay based on probed-associated dye fluorescence change. In a Strata-geneMx3005P Real-Time PCR System, the reactions were conducted according to the following protocol: the thermal cycling conditions were set at 95 °C for 10 min, followed by 40 cycles of 95 °C for 15 s and 60 °C for 1 min.

### 2.5. Determination of Serum β-Catenin Levels

An ELISA assay kit (MyBioSource Inc., San Diego, CA, USA, Catalogue No. MBS706120) was used in accordance with the manufacturer’s instructions to measure the levels of β catenin in serum samples from both patients and controls.

### 2.6. Statistical Analysis

The statistical analysis of the current was performed utilising V.22 of SPSS (Chicago, IL, USA). Mean ± Standard deviation (SD) was utilised for quantitative data, while qualitative data were displayed as percentages as well as numbers and percentages, and the Chi-squared test was conducted. To figure out the correlations between β catenin level and ITCH expression, Pearson’s correlation has been used. Survival analysis was evaluated by the Kaplan–Meier curve. At *p* ≤ 0.05, differences were found to be significant.

## 3. Results

### 3.1. Demographic Data and Clinical Findings of the Study Population

This study recruited a total of 124 participants, including 62 BC female patients and 62 healthy controls. Most variable distributions, including age and menopausal status, were comparable between BC cases and cancer-free controls (*p* value > 0.05). Regarding clinical findings, 71% of patients had positive LN metastasis, 64.5% were stage II + III, and 71.0% were grade II + III (Table 1).

### 3.2. The Relationship between the Circ-ICTH Expression and Clinico-Pathological Parameters of BC Patients

In order to assess the circ-ITCH expression in BC, the qRT-PCR analysis showed that circ-ITCH was considerably downregulated in BC compared to normal healthy tissues. In addition, cases showed lower expression with (0.64 ± 0.15) fold change compared to controls (Figure 2).

By analysing the association between circ-ITCH expression and BC clinical criteria (Table 2), low circ-ITCH expression was more significantly prone to lymph node metastasis (*p* = 0.003), larger tumour size (*p* = 0.04), advanced TNM stage (*p* = 0.004), and histological grade (*p* = 0.003). These outcomes suggest that circ-ITCH is a tumour suppressor gene and is essential for BC progression.

### 3.3. β-catenin Level (pg/mL) in Control and BC Patients and Its Correlation with Circ-ITCH mRNA Expression

Tumour development, progression, and metastasis are all accompanied by deregulated Wnt/catenin signalling in malignancies. Therefore, serum β-catenin (pg/mL) levels were tested in both the control and BC patients’ groups. The mean value of serum β-catenin in the BC patients’ group was statistically significantly higher than in the control group (*p* < 0.001) (Figure 3). Correlation between circ-ITCH mRNA expression and β-catenin protein level calculated on 62 breast cancer patients showed that there was a statistically significant negative correlation between circ-ITCH mRNA expression and β-catenin protein level in breast cancer patients (r − 0.3) (*p*-value < 0.05 *) (Figure 4).

### 3.4. Distribution of Genotype and Allelic Frequencies of Circ-ITCH Polymorphisms

Table 3 depicts the genotype distribution of the targeted SNPs and their relationships to the risk of BC. The genotype frequencies of SNPs were in line with the Hardy–Weinberg equilibrium (HWE) in both cases and controls (*p*-value > 0.05). According to the findings, the allele frequencies of rs10485505 C/T (71.0%/29% in BC vs. 87.1%/12.9% in control) (adjusted OR = 2.76; 95% CI = 1.38–5.59; *p*-value < 0.001) and rs4911154 G/A (67.7%/32.3% in BC vs. 83.9%/16.1% in control) (adjusted OR = 2.4; 95% CI = 1.29–4.77; *p*-value = 0.003) were significantly associated with BC risk. For rs10485505, compared with the genotype CC (54.8% in BC vs. 77.4%in control), genotype CT (32.3% in BC vs. 19.4% in control) (adjusted OR = 2.32; 95% CI = 1.0–5.94) and TT (12.9% in BC vs. 3.2% in control) (adjusted OR = 5.65; 95% CI = 1.01–41.9) were significantly related with increased BC risk; while for rs4911154, compared with the genotype GG (48.4% in BC vs. 71.0% in control), genotype GA (30.7% in BC vs. 25.8% in control) (adjusted OR = 2.2; 95% CI = 1.0–5.21) and AA (12.9% in BC vs. 3.2% in control) (adjusted OR = 5.87; 95% CI = 1.04–43.2) were strongly linked to an increased possibility of BC.

### 3.5. Association of rs10485505 Genotype and rs4911154 Genotype of Circ-ITCH Polymorphisms and Clinico-Pathological Features of BC Patients

In this study, we enrolled 62 cases of BC and divided them into groups based on their genotypes rs4911154 and rs10485505. Regarding the demographic characteristics, including age and menopausal state, no significant difference was observed. However, patients carrying the GA and AA genotypes of rs4911154 were significantly more likely to have larger tumour sizes, positive LN metastases, higher TNM stages, and histological grades. Regarding the rs10485505 genotype of circ-ITCH polymorphisms, our results proved that the carriers of the T allele frequently have poor prognosis compared to carriers of the CC genotype, as carriers of the CT and TT genotypes had significant positive LN metastasis, advanced tumour stage, and poor histological grade (*p* < 0.001) (Table 4).

### 3.6. Association of rs10485505 Genotype and rs4911154 Genotype of Circ-ITCH Polymorphisms, Circ-ITCH mRNA Expression and β Catenin Level in BC

We also examined the relative expression level of circ-ITCH mRNA in BC tissue samples obtained from various patients. To further examine the effect of the rs4911154 polymorphism, we also analysed the relative expression level of circ-ITCH mRNA in BC tissue samples derived from various patient groups. As depicted in Figure 5a, comparing individuals with the GG genotype of rs4911154 to those with the A allele, the relative expression levels of circ-ITCH in the BC samples of patients were significantly reduced. Therefore, it can be concluded that the A allele at rs4911154 is associated with the downregulation of circ-ITCH. In contrast, β-catenin protein levels in the serum of BC patients with the AA and GA genotypes increased significantly compared to those with the GG genotype of rs4911154 (Figure 5b). While circ-ITCH expression level and rs10485505 genotype carriers of the CT and TT genotypes showed downregulation of circ-ITCH expressions fold relative to the CC genotype, as depicted in Figure 5c. β-catenin levels were significantly increased in TT and CT genotypes compared to CC genotypes (*p* < 0.001) (Figure 5d).

### 3.7. The Evaluation of Circ-ITCH RNA Expression as a Prognosis Survival Biomarker in BC

To evaluate whether the expression of studied genes has an impact on patients’ clinical outcomes, disease-free survival (DFS) (Figure 6a) and overall survival (OS) (Figure 6b), Kaplan–Meier analyses were performed, and the results are reported in Figure 6. The follow-up periods for this study were 24 months. Patients were divided into two groups: low- and high-expression groups. Interestingly, patients with low circ-ITCH RNA expression had significantly longer DFS (*p* = 0.031) and OS (*p* = 0.005) than those with high expressions. The survival curves showed statically significant differences between the ‘low-expression’ and the ‘high-expression’ groups of circ-ITCH RNA expression. Additionally, logistic regression analysis showed that circ-ITCH rs10485505 (T allele), circ-ITCH rs4911154 (A allele), and low circ-ITCH RNA expression were the only factors that add significance to the model and considered factors predicting the prognosis of BC (*p* ≤ 0.05) (Table 5).

## 4. Discussion

As circ-ITCH shares some miRNA binding sites with the 3′UTR of ITCH and can regulate linear ITCH expression in different cancers [7], it is hypothesised that circ-ITCH is a crucial gene in BC development.

Compared to normal healthy tissues, circ-ITCH was significantly downregulated in BC, with a fold change of 0.64 ± 0.15 in this study. In addition, we discovered that low circ-ITCH expression was more strongly associated with lymph node metastasis, larger tumour size, advancing TNM stage, and histological grade. In accordance with our findings, Wang et al. hypothesised that circ-ITCH was downregulated in breast cancer, particularly in TNBC, compared to normal tissues, and low circ-ITCH expression was associated with lymph node metastasis, larger tumour size, advanced stages, and shorter survival times [19]. With respect to other types of cancers, Guo et al. suggested that increased expression of circ-ITCH was linked to a favourable survival of HCC and that HCC tissues had significantly lower levels of circ-ITCH expression than neighbouring tissues. They claimed that circ-ITCH is a susceptibility biomarker for HCC and has prognostic significance [20].

In prostate cancer tissues and PC cell lines, circ-ITCH expression levels were significantly lower than in adjacent normal tissues and human prostate epithelial cells, according to prostate cancer. In addition, circuit overexpression decreases tumour volume, slows tumour growth, and can inhibit malignant PC cell phenotypes [21]. The circ-ITCH regulates the Wnt/catenin signalling pathway in colorectal cancer cells, and its expression was approximately 75.6% higher in cancer-adjacent tissue than in cancer-matched tissue [7]. According to research by Peng et al., circ-ITCH is downregulated in gastric cancer and considered a prognostic marker, and by securing miR-17 via the Wnt/catenin pathway, circ-ITCH can prevent gastric cancer tumourigenesis [22]. Furthermore, circ-ITCH expression was downregulated in bladder cancer tissues and cell lines. Patients with BCa who had reduced circ-ITCH expression experienced shorter survival. Induced expression of circ-ITCH decreased cell proliferation, migration, invasion, and metastasis both in vitro and in vivo [23].

Two fundamental objectives of cancer research are identifying the critical pathways involved in the regulation of normal cell growth and understanding how cancer cells evade these regulatory systems. One of these pathways is the Wnt/catenin signalling pathway. During tumour development, growth, and metastasis, it is well known that malignancies exhibit dysregulated Wnt/catenin signalling [24]. Consequently, there is substantial interest in inhibiting catenin signalling as a cancer treatment [25].

Our study found that the serum level of β-catenin was significantly higher among BC patients compared to the healthy volunteer controls (*p* < 0.001). Correlation between β-catenin protein level and circ-ITCH mRNA expression showed a significant negative correlation. As a result of circ-ITCH’s role as a miRNA sponge, ITCH is produced at a higher level. ITCH is a component of the Wnt/catenin pathway and promotes the ubiquitination and degradation of phosphorylated Dvl2, thereby inhibiting canonical Wnt signalling [16].

Our findings supported those of an earlier study that examined the expression of the genes Znhit1 and HIF-2 in BC tissues and how these genes related to levels of the β-catenin in the tissues and serum of BC patients. They found that Znhit1 gene expression in BC tissues had a statistically significant negative correlation with β-catenin levels in the tissues and serum of BC patients. This finding can be attributed to the downregulation of Znhit1 in order to induce and maintain stemness in BC [26]. Another study revealed that β-catenin might effectively distinguish between patients with HCC and those with chronic hepatitis (CH), demonstrating that serum–catenin levels were significantly increased in patients with HCC compared to those with CH and healthy controls [27]. Moreover, serum *β*-catenin levels are gradually increased in colorectal polyps and CRC. However, there is no correlation between its levels and CRC disease development [28].

The fact that cytosolic β-catenin is a crucial mediator between stimulation of Wnt signalling and ensuing increases in Wnt1-inducible signalling pathway protein-1 (WISP-1) expression was one of the explanations for the underlying mechanism of catenin involvement in breast cancer [29,30]. WISP-1 expression is unnaturally elevated in pathological situations such as cancer and fibrosis [31]. WISP-1 has frequently been proposed as an oncogene in human breast cancer [32]. According to Chang et al., blood levels of β-catenin and WISP-1 were higher in breast cancer patients compared to controls, and they also discovered a significant positive connection between the two proteins in serum samples, also raising the prospect that circulating β-catenin could be employed as a breast cancer biomarker for prognosis and/or diagnosis [33].

CircRNAs can be used to determine the risk of developing cancer. A single DNA building block (nucleotide) of a gene or within the regulatory sections of a gene is referred to as a single nucleotide polymorphism (SNP). SNPs can be used as a biomarker to determine an individual’s cancer risk [34]. To our knowledge, no research has yet suggested a connection between the circ-ITCH gene polymorphisms rs10485505, rs4911154, and the risk of BC.

According to the findings of this study, the genotypes CT and TT of rs10485505 were significantly related to an increased risk of developing BC compared to the CC genotype. In contrast, the genotypes GA and AA of rs4911154 were significantly associated with an increased risk of developing BC compared to the GG genotype. No substantial difference was detected in the genotyping distribution of rs10485505 and rs4911154 concerning the fundamental features of BC patients, including age and menopausal status. However, patients with the GA and AA genotypes of rs4911154 had a significantly increased risk of larger tumours, positive LN metastasis, a more advanced TNM stage, and tumours with a higher histological grade. Regarding the circ-ITCH polymorphism genotype at rs10485505, carriers of the T allele typically have a poor prognosis (*p* < 0.001).

In prior studies, it was found that the circ-ITCH SNPs rs10485505 and rs4911154 were strongly related to increased hepatocellular carcinoma risk. These findings suggest that circRNAs may contain genetic HCC determinants. According to research by Guo et al., the circ-ITCH SNPs rs10485505 and rs4911154 are strongly linked to a higher risk of developing HCC. For rs10485505, genotype CT and TT were significantly associated with increased hepatocellular carcinoma risk compared to genotype CC, whereas for rs4911154, genotypes GA and AA were significantly associated with increased HCC risk compared to genotype GG (OR = 1.74; 95% CI = 1.21–2.49) [20]. These studies demonstrated that circ-ITCH and its genetic variation might be biomarkers for hepatocellular carcinoma prognosis and vulnerability [35].

We investigated the potential role of rs4911154 polymorphism and rs10485505 in BC, as well as their effect on the expression level of circ-ITCH mRNA in BC tissues in addition to b-catenin level in BC patients, the relative expression levels of circ-ITCH were remarkably decreased. In contrast, β-catenin serum levels increased significantly in the BC samples of patients carrying the AA and GA genotypes of rs4911154 compared to the GG genotype, whereas CT and TT genotype carriers exhibited downregulation on the circ-ITCH expression fold. In contrast, the β-catenin level increased significantly compared to the CC genotype (*p* < 0.001).

Similarly, Guo et al. claimed that single nucleotide polymorphisms of circ-ITCH’s rs4911154 could exacerbate the thyroid cancer’s malignant development into a thyroid nodule (TN). In addition, a signalling pathway including miR-22-3p, CBL, and circ-ITCH was constructed to explain how circ-ITCH SNPs affect thyroid tumour malignancy. Additionally, they evaluated how circ-ITCH, miR-22-3p, and Casitas B-lineage Lymphoma (CBL) were expressed in tissue samples of patients who had different rs4911154 alleles. The A allele in rs4911154 was related to the malignancy of thyroid nodules with decreased doubling in comparison to the G allele. Additionally, the A allele was related to increasing the expression of miR-22-3p and decreasing the expression of circ-ITCH/CBL. Additionally, the A allele was associated with decreased tissue apoptosis. When comparing thyroid nodule patients with the GG genotype to those with the GA and AA alleles, the expression of circ-ITCH and CBL was significantly upregulated, whereas the expression of miR-22-3p was significantly downregulated [36].

In this current study, survival analysis revealed that patients with higher circ-ITCH expression had better DFS and OS than patients with lower expression, and logistic regression analysis revealed that the only factors significantly contributing to the model and factors predicting the prognosis of BC were low circ-ITCH RNA expression and the circ-ITCH rs10485505 (T allele) and circ-ITCH rs4911154 (A allele). This suggested that circ-ITCH would be valuable as a prognostic predictor for BC; however, a greater carefully planned epidemiologic and functional inquiry is needed to confirm these results. According to a study that validates our findings, circ-ITCH expression was considerably lower in hepatocellular carcinoma (HCC) tissues than in neighbouring tissues, and circ-ITCH expression levels were higher in HCC tissues compared to adjacent tissues [20].

As an explanation for our findings and the potential role of rs10485505 and rs4911154 in circ-ITCH in increasing the risk of BC, Guo et al. found that because circ-ITCH may have an inhibitory effect on HCC, these SNPs in circ-ITHC were significantly associated with an increased risk of HCC [20]. Circ-ITCH enhanced the expression of its parental gene ITCH by acting as miRNA sponges for miR-7, miR-17, and miR-214, which suppressed the Wnt/catenin signalling pathway and prevented the proliferation of cancer cells [8,9,13,14,15]. This finding was also supported by Wang et al., who stated that in vitro circ-ITCH overexpression reduces papillary thyroid cancer cell invasion and proliferation while increasing apoptosis. According to the mechanism, miR-22-3p is sponged by circ-ITCH overexpression to increase CBL production, suppressing the Wnt/catenin pathway [37].

## 5. Conclusions

For the first time, a case–control study suggests that the circ-ITCH rs10485505 and rs4911154 SNPs influence breast cancer risk in Egyptian patients. Our research showed that the rs10485505 polymorphism (T allele) and the rs4911154 polymorphism (A allele) are related to the risk and prognosis of breast cancer. This may be attributed to their effect on the level of circ-ITCH mRNA expression in breast cancer tissues as well as the level of β-cateinin in BC patients. The relative expression levels of circ-ITCH were remarkably decreased on the contrary, β-catenin serum levels showed a significant markedly increase in the BC samples of patients carrying the A allele of rs4911154 and T allele of rs10485505 According to Kaplan–Meier survival analysis, the expression of circ-ITCH is associated with the prognosis of breast cancer. Furthermore, there is a correlation between the expression of circ-ITCH and tumour size, tumour grade, TNM stage, and clinical stage. Together, these findings point to a possible role for this biomarker in prognosis. This study conclusively establishes a connection between cancer susceptibility and circRNA SNPs.

Limitation: It is important to be aware of some study limitations. First off, the fact that this study solely takes into account the Egyptian population could restrict the generalizability of its conclusions to other populations. Second, the sample size was small; care should be taken when interpreting the data. Therefore, a bigger sample size is needed to assess the contribution of circ-ITCH polymorphisms to breast cancer risk. Third, other circ-ITCH polymorphisms and potential interactions were not examined. However, a variety of elements interact both singly and collectively to affect the risk of breast cancer. So, in our future work, we should consider more elements.

Recommendation: Further thorough investigations in a bigger sample are necessary to fully understand the complicated disease of breast cancer, which incorporates both genetic and environmental variables. These studies should look at the combined impact of numerous loci rather than just one polymorphic site or one gene. Circ-ITCH’s role in the aetiology of breast cancer is clearly apparent. To learn the reason for the elevated amount of circ-ITCH in breast cancer, more research must be conducted in addition to acknowledging it as a potential therapeutic target for breast malignancy.

## Figures and Tables

**Figure 1 diagnostics-13-02033-f001:**
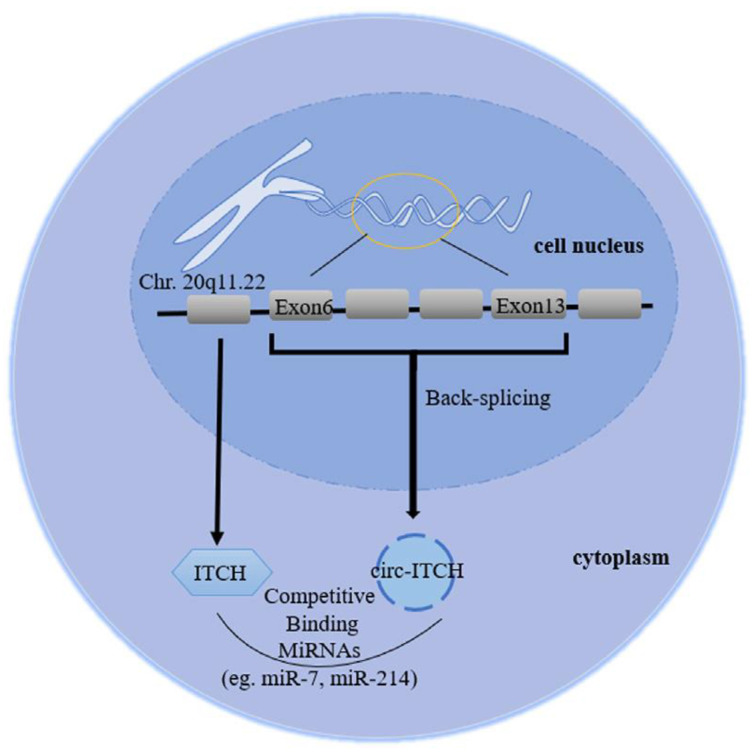
Diagram depicting the biogenesis of the circ-ITCH structure, the gene of which is found on chromosome 20q11.22 and was created by backsplicing from exons 6 to 13 of the ITCH coding gene [17].

**Figure 2 diagnostics-13-02033-f002:**
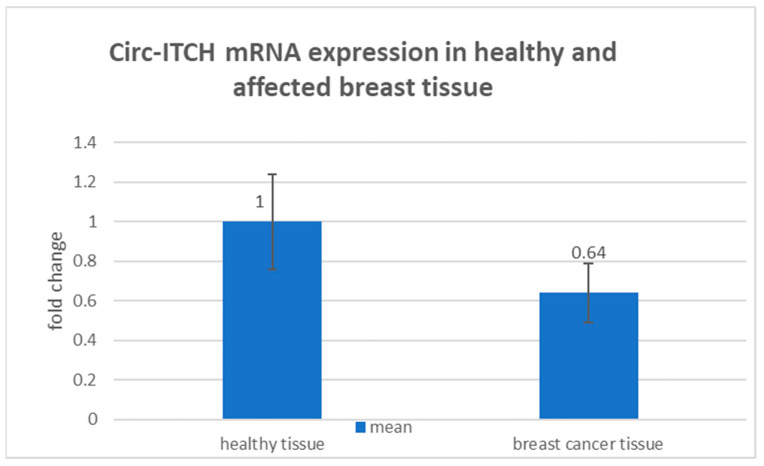
Bar chart representing the mean value of circ-ITCH RNA expression (fold change) in the control group and BC patients’ group. Expression level is in fold change; X ± SD for mean ± standard deviation; the mean value of circ-ITCH expression was significantly downregulated in breast cancer compared to normal healthy tissue. Cases showed lower expression with (0.64 ± 0.15) fold change compared to controls.

**Figure 3 diagnostics-13-02033-f003:**
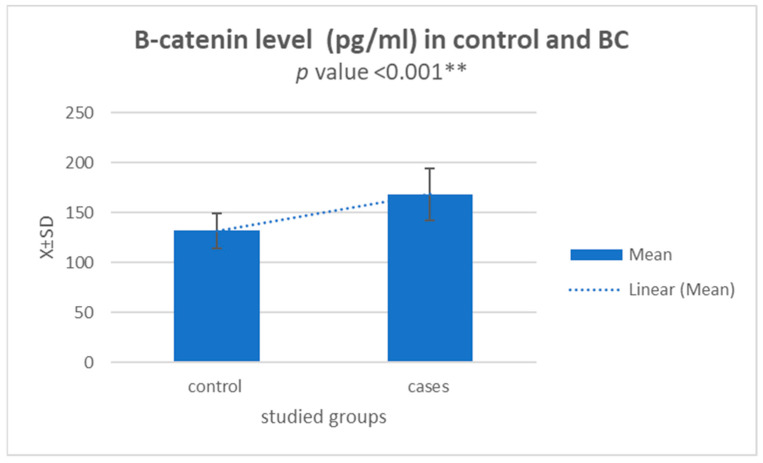
Bar chart representing the mean value of serum β-catenin (pg/mL) in the control group and BC patient group; the mean value of serum β-catenin in the breast cancer patients’ group was statistically significantly higher than that in the control group. ** statistically highly significant difference (*p* value < 0.001).

**Figure 4 diagnostics-13-02033-f004:**
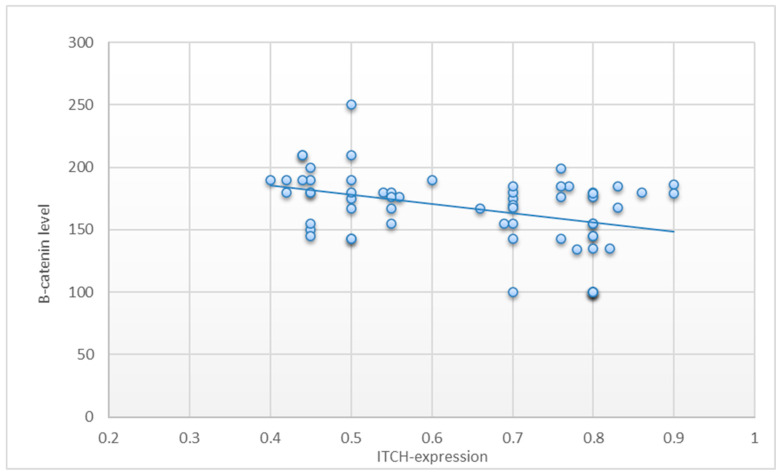
Pearson’s correlation calculated on 62 breast cancer patients showed that there was a statistically significant negative correlation between circ-ITCH mRNA expression and β-catenin protein level in breast cancer patients (r = 0.3) (*p*-value ≤ 0.05).

**Figure 5 diagnostics-13-02033-f005:**
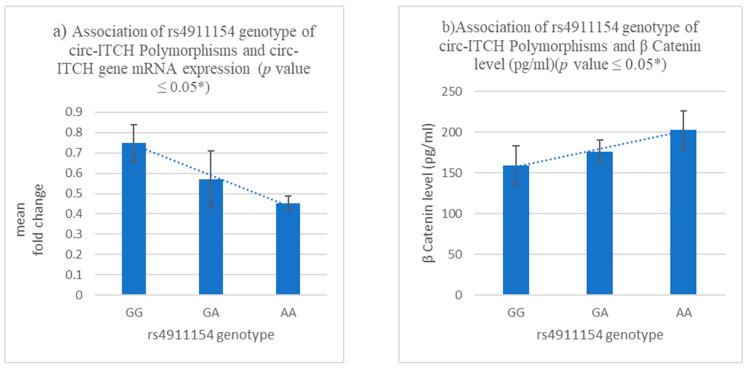
Bar chart shows the association of rs4911154 genotype of circ-ITCH polymorphisms with its mRNA expression fold change (**a**) and β-catenin protein level (**b**) and the association of rs10485505 genotype of circ-ITCH Polymorphisms with its mRNA expression fold change (**c**) and β-catenin protein level (**d**) *p*-value: * statistically significant difference (*p* ≤ 0.05).

**Figure 6 diagnostics-13-02033-f006:**
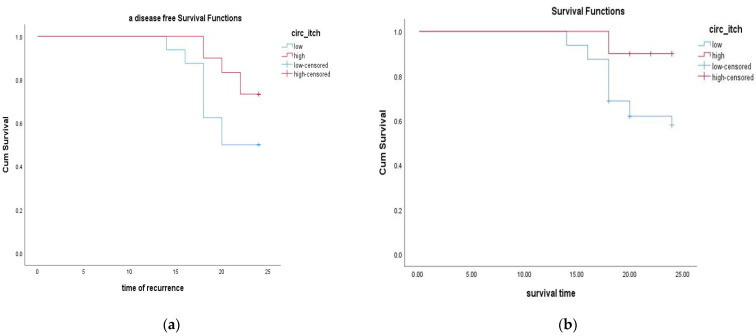
Kaplan–Meier analyses of disease-free survival (DFS) (**a**) and overall survival (OS) (**b**) correlated to circ-ITCH RNA expression (fold change) in breast cancer patients.

**Table 1 diagnostics-13-02033-t001:** Demographic data and clinical findings.

	Cases		Control		*p* Value
	N = 62	%	N = 62	%	
Age (years)					
≤40	6	9.7	8	12.9	0.57
>40	56	90.3	54	87.1	
Menopause					
No	20	32.3	26	41.9	0.26
Yes	42	67.7	36	58.1	
Tumour size (cm)					
≤2	31	50.0			
>2	31	50.0			
LN metastasis					
−ve	18	29.0			
+ve	44	71.0			
TNM stage					
I	22	35.5			
II + III	40	64.5			
Histological grade					
I	18	29			
II + III	44	71.0			

**Table 2 diagnostics-13-02033-t002:** Association of Circ-ITCH mRNA expression with clinico-pathological parameters in BC cases.

	Low Expression		High Expression		*p* Value	OR (95% CI)
	N	%	N	%		
Age (years)						
≤40	4	12.5	2	6.7	0.36	2.0 (0.28–17.3)
>40	28	87.5	28	93.3		
Menopause						
No	16	50.0	10	33.3	0.18	0.5 (0.16–1.57)
Yes	16	50.0	20	66.7		
Tumour size (cm)						
≤2	12	37.5	19	63.3	0.04 *	0.35 (0.11–1.1)
>2	20	62.5	11	36.7		
LN metastasis						
−ve	4	12.5	14	46.7	0.003 *	0.16 (0.04–0.66)
+ve	28	87.5	16	53.3		
TNM stage						
I	6	12.5	16	53.3	0.004 *	0.2 (0.05–0.72)
II + III	26	81.2	14	46.7		
Histological grade						
I	4	12.5	14	46.7	0.003 *	0.16 (0.04–0.66)
II + III	28	87.5	16	53.3		

Median circ-ITCH values were used as a cutoff, *p*-value * statistically significant difference (*p* ≤ 0.05).

**Table 3 diagnostics-13-02033-t003:** Distribution of genotype and allelic frequencies of circ-ITCH polymorphisms and the risk of breast cancer.

	Cases		Control		OR (95% CI)	*p*-Value
	N = 62	%	N = 62	%		
rs10485505						
Genotype						
CC	34	54.8	48	77.4	1.0	
CT	20	32.3	12	19.4	2.32 (1.0–5.94)	0.04 *
TT	8	12.9	2	3.2	5.65 (1.01–41.9)	0.02 *
Alleles						
C	88	71.0	108	87.1	1.0	
T	36	29.0	16	12.9	2.76 (1.38–5.59)	<0.001 **
rs4911154						
Genotype						
GG	30	48.4	44	71.0	1.0	
GA	24	30.7	16	25.8	2.2 (1.0–5.21)	0.04 *
AA	8	12.9	2	3.2	5.87 (1.04–43.2)	0.01 *
Alleles						
G	84	67.7	104	83.9	1.0	
A	40	32.3	20	16.1	2.4 (1.29–4.77)	0.003 *

*p*-value: * statistically significant difference (*p* ≤ 0.05), ** statistically highly significant difference (*p* ≤ 0.001).

**Table 4 diagnostics-13-02033-t004:** Association of rs10485505 genotype and rs4911154 genotype of circ-ITCH polymorphisms and clinico-pathological features of BC patients.

	rs10485505 Genotype							rs4911154 Genotype						
	CC		CT		TT		*p*-Value	GG		GA		*AA*		*p*-Value
	N = 34	%	N = 20	%	N = 8	%		N = 30	%	N = 24	%	N = 8	%	
Age (years)														
≤40	4	11.8	2	10.0	0	0.0		4	13.3	2	8.3	0	0.0	
>40	30	88.2	18	90.0	8	100.0	0.59	26	86.7	22	91.7	8	100.0	0.5
Menopause														
No	12	35.3	10	50.0	4	50.0	0.5	10	33.3	13	54.2	3	37.5	0.29
Yes	22	64.7	10	50.0	4	50.0		20	66.7	11	45.8	5	62.5	
Tumour size (cm)														
≤2	27	79.4	3	15	1	12.5	<0.001 **	25	83.3	6	25.0	0	0.0	<0.001 **
>2	7	20.6	17	85	7	87.5		5	16.7	18	75.0	8	100.0	
LN metastasis														
−ve	17	50.0	1	5.0	0	0.0	<0.001 **	15	50	3	12.5	0	0.0	
+ve	17	50.0	19	95.0	8	100.0		15	50	21	87.5	8	100.0	0.0016 *
TNM stage														
I	16	47.1	1	5.0	1	12.5		20	66.7	2	8.3	0	0.0	
II + III	18	52.9	19	95.0	7	87.5	<0.001 *	10	33.3	22	91.7	8	100.0	<0.001 **
Histological grade														
I	16	47.1	1	5.0	1	12.5		17	56.7	1	4.2	0	0.0	
II + III	18	52.9	19	95.0	7	87.5	<0.001 *	13	43.3	23	95.8	8	100.0	<0.001 **

*p*-value * statistically significant difference (*p* ≤ 0.05), ** statistically highly significant difference (*p* ≤ 0.001).

**Table 5 diagnostics-13-02033-t005:** Univariate and multivariate logistic regression analysis for prediction of prognosis of BC.

	Univariate Analysis: OR (95% CI)	Multivariate Analysis: OR (95% CI)
Age > 40 years	1.1 (0.2–3.2)	-------------------
Menopause	0.5 (0.17–1.3)	--------------------
Tumour size > II	1.59 (0.63–4.05)	--------------------
LN metastasis	2.54 (1.06–6.16) *	1.71 (0.85–2.46)
TNM stage II–III	1.89 (0.88–4.09)	--------------------
Histological grade II–III	1.42 (0.64–3.15)	--------------------
circ-ITCH rs 10485505 (T allele)	2.2 (1.04–4.75) *	2.15 (1.09–4.3) *
circ-ITCH rs 4911154 (A allele)	2.43 (1.15–5.17) *	2.34 (1.19–4.6) *
low circ-ITCH RNA expression	2.67 (1.27–5.65) *	2.56 (1.3–4.07) *

*p*-value * statistically significant difference (*p* ≤ 0.05).

## Data Availability

Data are available on request due to privacy/ethical restrictions.

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
