# Peer review of "Role of Circ-ITCH Gene Polymorphisms and Its Expression in Breast Cancer Susceptibility and Prognosis"

_diagnostics, 2023, doi:10.3390/diagnostics13122033_

Round 1

Reviewer 1 Report

 The authors explored the association between cir-ITCH gene polymorphisms, expression, and their effect on β-catenin levels correlating this with the development of breast 14 cancer. The current version of the manuscript could be improved by following aspects:

  1. Based on the author's findings, the β-catenin level is increased in breast cancer patients. However, the importance of β-catenin level is not well explained throughout the manuscript and it should be included.
  2. The results and discussion section should be improved. For example, why the β-catenin level is high in BC samples and what would be underlying mechanism is explained clearly.
  3. It is interesting to know how the rs10485505 and rs4911154 genotype is important over other variants/mutations. What is the frequency of these mutations in the BC sample and other samples?
  4. All the figures, the x-axis and y-axis title is not clear and it would be improved
  5. The details of the statistical test should be provided
  6. The structure of Cir-ITCH gene should be provided as a figure

Author Response

Dear Editor,

Thank you for giving us the opportunity to revise and resubmit this manuscript to the Diagnostics journal. We would like to thank the reviewers and the editor for careful and thorough reading of this manuscript and for the thoughtful comments and constructive suggestions, which help to improve the quality of this manuscript. Please, allow us to explain our responses.

Reviewer 1#

Comment 1:-

  1. Based on the author's findings, the β-catenin level is increased in breast cancer patients. However, the importance of β-catenin level is not well explained throughout the manuscript and it should be included.

 Response

This was done and highlighted in the manuscript

Comment 2:-

  1. The results and discussion section should be improved. For example, why the β-catenin level is high in BC samples and what would be underlying mechanism is explained clearly.

Response

This was done and highlighted in the manuscript

Comment 3:-

  1. It is interesting to know how the rs10485505 and rs4911154 genotype is important over other variants/mutations. What is the frequency of these mutations in the BC sample and other samples?

Response

According to the findings, the allele frequencies of rs10485505 C/T (71.0%/29% in BC vs 87.1%/12.9% in control) (adjusted OR =2.76; 95% CI=1.38-5.59; P value <0.001) and rs4911154 G/A (67.7%/32.3% in BC vs 83.9%/16.1% in control) (adjusted OR =2.4; 95% CI=1.29-4.77; P value =0.003) were significantly associated with BC risk. For rs10485505, compared with the genotype CC (54.8% in BC vs 77.4%in control), genotype CT(32.3%in BC vs 19.4%in control)( (adjusted OR =2.32; 95% CI=1.0-5.94) and TT(12.9% in BC vs 3.2% in control) (adjusted OR =5.65; 95% CI= 1.01-41.9) were significantly related with increased BC risk; while for rs4911154, compared with the genotype GG(48.4%in BC vs 71.0% in control), genotype GA(30.7% in BC vs 25.8% in control) (adjusted OR =2.2; 95% CI= 1.0-5.21) and AA(12.9% in BC vs 3.2% in control) (adjusted OR =5.87; 95% CI= 1.04-43.2) were strongly linked to an increased possibility of BC.

This was mentioned in table three and added in the comment in the result section

Comment 4:-

  1. All the figures, the x-axis and y-axis title is not clear and it would be improved

Response

This was done

Comment 5:-

  1. The details of the statistical test should be provided

Response

This was added in the statistical analysis section and highlighted

Comment 6:-

  1. The structure of Cir-ITCH gene should be provided as a figure

Response

This was done

Reviewer 2 Report

The study is very informative and can be the potential target in the future drug discovery and development. however, the statistical calculations should be revised and cross checked including p-value and Pearson`s correlation calculated, whether the results are significant or not. 

Which statistical test were used. it should be mentioned in the text. 

Moreover, the discussion should be improved by adding more references.

Abstract and conclusion part require attention. 

Manuscript may be accepted after minor issues

The manuscript is well written. only in introduction part few errors were found. author must revise whole manuscript before revision submission. 

Author Response

Reviewer 2#

Comment 1:-

The study is very informative and can be the potential target in the future drug discovery and development. however, the statistical calculations should be revised and cross checked including p-value and Pearson`s correlation calculated, whether the results are significant or not. 

Response 1:-

This was done

Comment 2:-

Which statistical test were used. it should be mentioned in the text. 

Response 2:-

This was added in the statistical analysis section and highlighted

Comment 3:-

Moreover, the discussion should be improved by adding more references.

Response 3:-

This was done

Comment 4:-

Abstract and conclusion part require attention. 

Response 4:-

This was done

Comment 5:-

Comments on the Quality of English Language

The manuscript is well written. only in introduction part few errors were found. author must revise whole manuscript before revision submission. 

Response 5:-

This was done

Reviewer 3 Report

(1) Consider including a brief introduction to provide an overview of breast cancer, emphasizing the need for identifying genetic markers and prognostic indicators.

(2) Provide a clear explanation of the Circ-ITCH gene and its biological relevance to breast cancer pathogenesis in order to establish a solid foundation for the study, the provided one is insufficient.   (3) Figure ledges need to be made more clear and in size, some are overlapping.  
(4) Include a comprehensive description of the methodology used to identify and analyze Circ-ITCH gene polymorphisms, as well as the techniques employed to measure its expression levels.
(5) Consider presenting the demographic and clinical characteristics of the study population, including age, gender, tumor stage, and other relevant factors, to ensure the readers have a complete understanding of the cohort.   (6) Provide detailed statistical analyses to support the associations between specific Circ-ITCH gene polymorphisms and breast cancer susceptibility, incorporating p-values and confidence intervals where applicable.   (7) Include a graphical representation, such as a Kaplan-Meier survival curve, to visually demonstrate the impact of Circ-ITCH expression on breast cancer prognosis, if any.   (8) Consider discussing the potential limitations of the study, such as the sample size, and address any potential confounding factors that might have influenced the results.
(9) Suggest potential future directions for research, such as investigating the functional implications of specific Circ-ITCH gene polymorphisms and conducting validation studies in larger cohorts.
(10) Discuss the clinical implications of the findings, emphasizing how Circ-ITCH expression and genetic markers could be integrated into current diagnostic and prognostic approaches for breast cancer patients.
(11) Consider revising the conclusion to succinctly summarize the key findings and their significance, highlighting the potential impact of Circ-ITCH gene polymorphisms and expression on breast cancer susceptibility and prognosis.   (12) The article contains grammatical and logical errors and typos; consider improving it to increase readability.

Moderate editing of English language

Author Response

Dear Editor,

Thank you for giving us the opportunity to revise and resubmit this manuscript to the Diagnostics journal. We would like to thank the reviewers and the editor for careful and thorough reading of this manuscript and for the thoughtful comments and constructive suggestions, which help to improve the quality of this manuscript. Please, allow us to explain our responses.

Reviewer 3#

Comment 1:-

(1) Consider including a brief introduction to provide an overview of breast cancer, emphasizing the need for identifying genetic markers and prognostic indicators.

Response1: this was done

Comment2:

(2) Provide a clear explanation of the Circ-ITCH gene and its biological relevance to breast cancer pathogenesis in order to establish a solid foundation for the study, the provided one is insufficient.  

Response

This was done

Comment 3:-

(3) Figure ledges need to be made more clear and in size, some are overlapping.  

Response

This was done

Comment 4:-

(4) Include a comprehensive description of the methodology used to identify and analyze Circ-ITCH gene polymorphisms, as well as the techniques employed to measure its expression levels.
Response

This was done and highlighted in the manuscript

Comment 5:

(5) Consider presenting the demographic and clinical characteristics of the study population, including age, gender, tumor stage, and other relevant factors, to ensure the readers have a complete understanding of the cohort.  

Response

This was presented in table 1

Comment 6:

(6) Provide detailed statistical analyses to support the associations between specific Circ-ITCH gene polymorphisms and breast cancer susceptibility, incorporating p-values and confidence intervals where applicable.  

Response

This was done and highlighted in the manuscript

Comment 7:

 (7) Include a graphical representation, such as a Kaplan-Meier survival curve, to visually demonstrate the impact of Circ-ITCH expression on breast cancer prognosis, if any.  

Response

This was done

Comment 8:

 (8) Consider discussing the potential limitations of the study, such as the sample size, and address any potential confounding factors that might have influenced the results.
Response

This was done and highlighted in the manuscript

Comment 9:

(9) Suggest potential future directions for research, such as investigating the functional implications of specific Circ-ITCH gene polymorphisms and conducting validation studies in larger cohorts.
Response

This was done and highlighted in the manuscript

Comment 10:

(10) Discuss the clinical implications of the findings, emphasizing how Circ-ITCH expression and genetic markers could be integrated into current diagnostic and prognostic approaches for breast cancerpatients.
Response

This was done and highlighted in the manuscript

Comment 11:

(11) Consider revising the conclusion to succinctly summarize the key findings and their significance, highlighting the potential impact of Circ-ITCH gene polymorphisms and expression on breast cancer susceptibility and prognosis.  

Response

This was done and highlighted in the manuscript

Comment 12:

 (12) The article contains grammatical and logical errors and typos; consider improving it to increase readability.

 Response

This was done

Sincerely,

Round 2

Reviewer 1 Report

I appreciate the author's answer, and the manuscript has significantly improved.

Reviewer 3 Report

The authors have carefully addressed by comments and improved the quality of the paper. It is now suitable for publication.

 Minor editing of English language required